



# 1 Size resolved morphological properties of the high

# 2 Arctic summer aerosol during ASCOS-2008

**E. Hamacher-Barth[1], C. Leck[1] and K. Jansson[2]**
[1] {Department of Meteorology, Stockholm University, Stockholm, Sweden}
[2] {Department of Materials and Environmental Chemistry, Stockholm
University, Stockholm, Sweden}
Correspondence to: E. Hamacher-Barth (evelyne@misu.su.se)
**Abstract**
The representation of aerosol properties and processes in climate models is fraught
with large uncertainties. Especially at high northern latitudes a strong under-
prediction of aerosol concentrations and nucleation events is observed and can only be
constrained by in situ observations based on the analysis of individual aerosol
particles. To further reduce the uncertainties surrounding aerosol properties and their
potential role as cloud condensation nuclei this study provides observational data
resolved over size on morphological and chemical properties of aerosol particles
collected in the summer high Arctic, north of 80° N.
Aerosol particles were imaged with scanning and transmission electron microscopy
and further evaluated with digital image analysis. In total 3903 particles were imaged
and categorized according to morphological similarities into three gross
morphological groups, single particles, gel particles and halo particles. Single
particles were observed between 15 nm and 800 nm in diameter and represent the
dominating type of particles (82%). The majority of particles appeared to be marine
gels with a broad Aitken mode peaking at 70 nm accompanied by a minor fraction of
ammonium (bi)sulfate with a maximum in number concentration at 170 nm. Gel
particles (11% of all particles) were observed between 45 nm and 800 nm with a
maximum in number concentration at 154 nm. Imaging with transmission electron
microscopy allowed further morphological discrimination of gel particles in
*"aggregate"* particles, *"aggregate with film"* particles and *"mucus-like"* particles.



Halo particles were observed above 75 nm and appeared to be ammonium (bi)sulfate
(59% of halo particles), gel matter (19%) or decomposed gel matter (22%) internally
mixed with sulfuric acid and/or methane sulfonic acid or ammonium (bi)sulfate with a
maximum in number concentration at 161 nm in diameter.
Elemental dispersive X-ray spectroscopy analysis of individual particles revealed
prevalence of the monovalent ions $Na^+/K^+$ for single particles and *"aggregate"*
particles and of the divalent ions $Ca^{2+}/Mg^{2+}$ for *"aggregate with film"* particles and
"mucus-like" particles. Emanating from those results and in agreement with model
studies reported elsewhere we propose a relationship between the availability of a
$Na^+/K^+$ and $Ca^{2+}/Mg^{2+}$ and the length of the biopolymer molecules participating in the
formation of the 3D gel networks.
**1  Introduction**
Aerosol particles have major impacts on the climate of our planet. On average they
increase the planetary albedo both directly by absorbing and scattering sunlight and
indirectly by modifying the reflectivity, life-time and extent of clouds (Twomey,
1977; Albrecht, 1989; Solomon et al., 2007). Despite of the shown importance aerosol
particles have on clouds their effects still give rise to large uncertainties in climate
models (Schimel et al., 1996; Penner et al., 2001; Forster et al., 2007). Detailed model
analyses have contributed to an enhanced understanding of the parametric
uncertainties in global aerosol models and point towards significant uncertainties
arising from an incomplete representation of aerosol processes and emissions in the
models (e.g. Lee et al., 2013; Carslaw et al., 2013; Mann et al., 2014). Especially for
high northern latitudes a strong under-prediction of aerosol particle concentrations
and nucleation events in summer compared to measurements is recognized (Mann et
al., 2014) leading to an insufficient representation of cloud condensation nuclei
(CCN), the fraction of an aerosol particle population that can activate and form cloud
droplets, within the models. Moreover, the microphysical properties of the cloud
droplets are strongly related to the size, chemical composition, morphology and state
of mixture of the activated CCN. Size resolved data based on the analysis of
individual particles are therefore indispensible for an appropriate parameterization of
aerosol particles within aerosol models.
The sources of aerosol particles in the Arctic are subjected to large regional and
seasonal differences. In late winter/spring a pronounced anthropogenic influence on





1 the Arctic is observed, a phenomenon known as Arctic haze (Shaw, 1995). During

2 that time the Arctic air mass expands southwards towards Eurasia and North America

3 and anthropogenic emissions are transported into the Arctic where they remain for

4 prolonged times (Shaw, 1995; Douglas and Sturm, 2003). The aerosol during periods

5 of Arctic haze is characterised by relatively high concentrations of aged aerosol

6 predominantly in the accumulation mode (Shaw, 1984; Heintzenberg and Leck, 1994;

7 Ström et al., 2003; Engvall et al., 2008; Korhonen et al., 2008). The air masses

8 arriving in summer, however, originate from sectors over the oceans with limited

9 man- made activities and the transport into the Arctic is slower compared to winter

10 conditions (Stohl, 2006). The summer conditions are thus much more pristine and the

11 aerosol shifts from being accumulation mode dominated to be Aitken mode

12 dominated (Heintzenberg et al., 2006; 2015; Engvall et al., 2008).

13 Over the high Arctic pack ice north of 80° number concentrations of CCN show a

14 large temporal variability, ranging over 2-3 orders of magnitude but usually are below

15 $100 \ cm^{-3}$ and occasionally less than $1 \ cm^{-3}$ (Lannefors et al., 1983; Bigg et al., 1996;

16 Bigg and Leck, 2001a; Mauritsen et al., 2011; Leck and Svensson, 2015). These

17 relatively low CCN concentrations have a significant impact on the formation of low-

18 level (stratiform) clouds prevalent in the high Arctic summer. Mauritsen et al. (2011)

19 identified a regime with very low CCN concentrations ($< 10 \ cm^{-3}$) where cloud

20 formation is limited mainly by the availability of CCN. Such low CCN concentrations

21 occur as a result of weak local aerosol sources and effective wet deposition (Nilsson

22 and Leck, 2002; Held et al., 2011a,b; Heintzenberg et al., 2006; Leck and Svensson,

23 2015) at the marginal ice zome and over the pack ice.

24 However, the physical and chemical properties which determine the ability of the

25 summer high Arctic aerosol particles to act as CCN are still not very well understood.

26 Attempts to theoretically predict concentrations of CCN in closure studies resulted in

27 both over- and under-predictions of the observed CCN concentrations (Zhou et al.,

28 2001; Bigg and Leck, 2001a; Lohman and Leck, 2005; Martin et al., 2011; Leck and

29 Svensson, 2015). The most recent closure study by Leck and Svensson (2015)

30 simulated the cloud nucleation process by assuming Köhler theory together with a

31 Lagrangian adiabatic air parcel model that solves the kinetic formulation for

32 condensation of water on size resolved aerosol particles. The authors suggested a

33 larger fraction of the internally/externally mixed water-insoluble particles in the

34 smaller aerosol size ranges and kinetically restricted growth of the activated particles.





The non-water soluble particle fraction was suggested to physically and chemically
behave as polymer gels[1] with a dichotomous behavior (low hygroscopic growth factor
but a high CCN activation efficiency) in cloud droplet activation as a result of the
interaction of the hydrophilic and hydrophobic entities on the structures of the high
Arctic polymer gels (Orellana et al., 2011). On average 32% of the Arctic DOM
assembled as microgels (Orellana et al., 2011), a significantly higher percentage than
reported for other ocean regions (10%; Chin et al., 1998; Verdugo et al., 2004). All
together these findings strongly supported the previously unverified hypothesis of a
link between cloud formation and polymer gels in the surface microlayer (SML,
<1000 μm thick at the air-sea interface) of the high Arctic open leads (Bigg et al.,
2004; Leck and Bigg, 1999; Leck et al., 2002; Leck and Bigg, 2005b; Bigg and Leck,
2008; Leck and Bigg, 2010).
The transport of marine gels into the atmosphere is thought to happen via the burst of
air bubbles at air-sea interface. Air bubbles rising within the water column scavenge
surface-active organic matter especially from the surface microlayer to their outer
walls (Wotton and Preston, 2005). Bursting of the bubbles at the water surface
produces small jet and film drops containing organic surface-active compounds,
debris of phytoplankton, bacteria, viruses and sea salt (Blanchard and Woodcock,
1957; Blanchard, 1971, Blanchard and Syzdek, 1988; Gershey, 1983; O'Dowd et al.,
1999) which are transported further up into the atmosphere through turbulent mixing
processes. However, studies of individual particles by Bigg and Leck (2001; 2008);
Leck et al., (2002); Leck and Bigg (2005a; 2005b) over the perennial ice have failed
to find evidence of sea salt particles of less than 200 nm in diameter. In the Arctic
breaking waves as a source of bubble are rare due to low wind speeds and short
fetches between the ice floes (Tjernström et al., 2012). Even in the absence of wind-
driven breaking waves a recent study has now confirmed both the presence and
temporal variability of a population of bubbles within the open leads (Norris et al.,
2011). The considered mechanisms for bubble formation and mixing were induced by
changes in gas saturation. Other possible bubble formation mechanisms are
respiration from algae and phytoplankton (Medwin, 1970; Johnson and Wangersky,
1987) and the release of trapped air from melting ice (Wettlaufer, 1998).

---

[1] Phytoplankton and bacteria in surface seawater produce varying amounts of mucus- or gel-like matter comprised of biopolymers like proteins, polysaccharides or lipids that form 3-dimensional networks inter-bridged with divalent ions. This type of supramolecular organisation is referred to as marine gels. (Verdugo, 2012 gives a review).



Due to the remoteness and the harsh conditions in the high Arctic the number of
aerosol studies from this region is limited; the data available rely on four expeditions
onboard the Swedish icebreaker *Oden* during the summers of 1991, 1996, 2001 and
2008 (Leck et al., 1996; 2001, 2004; Tjernström et al., 2014). These expeditions took
advantage of the pristine conditions during the Arctic summer when the Arctic is to a
great extent separated from air masses from polluted mid-latitudinal sources which
provided the unique opportunity to study aerosol particles from predominantly natural
sources. All other Arctic studies on aerosol chemical composition, morphology and
state of mixture were either performed during winter/spring when transport of
polluted aerosol from lower latitudes into the high Arctic is strong (e.g. Hara et al.,
2003; Xie et al., 2007; Winiger et al., 2015) and/or they were located further south,
outside the pack ice area, missing potential aerosol sources from the pack-ice area and
at the same time including anthropogenic pollutions, e.g. studies from Svalbard (Geng
et al., 2010; Chi et al., 2015).
To further reduce the uncertainties surrounding the CCN properties that promote
/suppress cloud droplet formation over the pack ice area an investigation of size,
chemical composition, morphology and state of mixture on the level of individual
aerosol particles is required. The present study will make use of aerosol particles
collected during the most recent icebreaker expedition under the name ASCOS
(Arctic Summer Cloud and Ocean Study) 2008. The Swedish icebreaker *Oden*
departed from Longyearbyen on Swalboard on 2 August and returned on 9 September
2008. After traversing the pack-ice northward the icebreaker was moored to an ice
floe and drifted passively with it around 87° N between 12 August and 1 September
(Tjernström et al., 2014). We used electron microscopy (scanning electron
microscopy (SEM) and transmission electron microscopy (TEM)) to image aerosol
particles at high resolution, and subsequent digital image analysis to objectively
assess size and morphology of the particles on an individual basis. Earlier studies
north of 80° focused rather on a qualitative description of the aerosol in the high
Arctic (Leck and Bigg, 2005, 2008, 2010; Bigg and Leck, 2001b, 2008)
complemented with bulk chemical analyses (Leck et al., 2002; 2013; Leck and
Svensson, 2015; Lohman and Leck, 2005).
By individually screening close to 4000 aerosol particles collected during the ice-drift
with SEM and subsequent digital mapping we firstly gained size resolved information
on the aerosol population as a whole. The obtained number size distribution was



compared with measurements from an independent method (Tandem Differential
Mobility Particle Sizer, TDMPS) to verify that a representative fraction of the aerosol
population was captured with SEM. Secondly, we sorted all mapped particles
according to morphological differences. A separate number size distribution for each
of the morphological types was obtained. Thirdly, to obtain deeper insights into the
morphological features of the collected particles and to simultaneously assess their
elemental composition with EDX spectroscopy we investigated a subpopulation of
aerosol particles in TEM at very high resolution.



**2   Methods for sampling of airborne aerosol particles during ASCOS**
**2.1   Collection of airborne particles**
**2.1.1   The sampling inlet**
A $PM_{10}$-inlet (9 cm inner diameter) was deployed at ambient conditions (85 – 100 %
relative humidity (RH) and temperatures around 0 °C) to eliminate particles with
diameters > 10 micrometer in equivalent aerosol dynamic diameter (EAD) from the
sampled air. To optimise the distance from the sea surface and the ship's
superstructure the inlet was located forward ~ 25 m above the sea surface and 3 m
above the roof of the laboratory container on the 4th deck of the icebreaker. Direct
contamination from the ship was excluded by using a pollution controller, located
directly after the inlet pipe that passed through the roof of the container. Provided that
the wind was within ± 70° of the direction of the bow and stronger than 2 ms$^{-1}$, no
pollution reached the sample inlets (Leck et al., 1996). Directly downstream from the
pollution sensor the electrostatic precipitator and the TDMPS were connected to the
inlet with short stainless steel tubes (length ca. 1 m). To ensure that the sampling
conditions and losses were the same for both instruments, the inlet take-offs for the
two samplers were placed closely together. The temperature in the container was kept
at 20 °C which resulted in a RH of 20% in the secondary lines during sampling. See
Leck et al. (2001) for more details of the set-up for the sampling of aerosol particles.
Volatile compounds on particle surfaces and weakly bound water molecules were
probably lost during the sampling procedure. In the Arctic the concentration of
volatile compounds is generally lower than at lower latitudes (Bates et al., 1987) and
losses due to evaporation can thus be considered very small.
**2.1.2   The electrostatic precipitator**
Using the electrostatic precipitator the aerosol particles were collected directly onto
3 mm copper 300 mesh Formvar-coated TEM grids (TED PELLA Inc.; Dixkens and
Fissan, 1999; Leck and Bigg, 2008). Formvar-coated grids were chosen because of
the hydrophilic and thus polar nature of the Formvar film (Rocha et al., 2005). The
aerosol particles were charged at the inlet of the precipitator by a $^{63}$Ni beta-emitting
radioactive source and precipitated by a 12 kV cm$^{-1}$ electric field between the inlet
and the collecting grid surface. The flow rate was kept very low (0.17 mL s$^{-1}$) in order
to collect particles up to ~1 μm in diameter. The collection efficiency of the


electrostatic precipitator was intercompared with a TSI 1236 Nanometer Aerosol
Sampler ($^{63}$Ni beta-emitting radioactive source and sample flow of 1 Lm$^{-1}$) mounted
side-by-side with the electrostatic precipitator. Both collected a small, but statistically
significant number of particles < 25 nm in diameter. The precipitator took samples for
6 to 12 hrs. Before and after sampling the grids were placed within a grid holder box
in a sealed plastic bag, together with silica gel packets, and stored in a desiccator at a
constant temperature of 20 °C in a clean room before they were investigated.
**2.1.3  The TDMPS-sampling system**
The TDMPS-sampling system to measure the number size distributions of dry (20%
RH) sub-micrometer particles used pairs of differential mobility analyzers (DMAs).
The TSI 3010 counters in the DMAs were size and concentration calibrated against an
electrometer and the TSI 3025 counters for particle sizes below 20 nm diameter
according to Stolzenburg (1988). This set up yielded a complete number size
distribution from 3 nm to 800 nm diameter scanned over 45 size channels every 10 -
20 min. Further details of the TDMPS system can be found in Heintzenberg and Leck
(2012). NIST (National Institute of Standards Technology) traceable calibration
standards of polystyrene latex spherical particles were used to determine error in
determination of the mobility diameter to ± 5 % (Wiedensohler et al., 2012).
In order to compare the number size distribution obtained from the precipitator
samples (section 2.1.2) with those simultaneously recorded by the TDMPS we
assumed median particle number diameters for each of the 45 TDMPS size channels.
The particle diameters were then merged to form a complete set of diameters across
the TDMPS measuring interval.
**2.2  Image recording and elemental analysis**
**2.2.1  Imaging with scanning electron microscopy (SEM)**
The samples were investigated with a high-resolution SEM (JEOL JSM-7401) under
high vacuum conditions, at less than 9.63 x 10-5 Pa (Stevens et al., 2009). A detailed
description of the setup of the scanning electron microscope can be found in
Hamacher-Barth et al. (2013). The Gentle Beam mode of the microscope was used to
minimize radiation damage of the aerosol particles, avoid surface charge-up and to
demagnify the electron beam diameter (Michael et al., 2010). Correction for
stigmatism and focusing of the electron beam was done every time before imaging an



aerosol particle. The grey scale (contrast and brightness) was adjusted automatically
before recording an image.
The imaging of the aerosol particles aimed to account for an uneven distribution of
the particles on the TEM grid and to capture a representative fraction of the aerosol
particles. In brief, particles were imaged at a magnification of 40.000 on the TEM
grid squares, along a diagonal from the center of the grid to the edge on 6 to 8 squares
of the TEM grid. Each square was screened systematically to capture a representative
fraction of the aerosol population. For a detailed description of the screening
procedure see Hamacher-Barth et al. (2013).
**2.2.2  Imaging with transmission electron microscopy (TEM)**
To image the samples with TEM they have to be coated by a thin metal layer.
Evaporation of a heavy metal thin coating at an oblique angle onto the sample
increases the mass contrast and accentuates the topography of the aerosol particle by
producing a shadow (William and Carter, 2006). Furthermore shading has the
advantage that the metal cover protects the aerosol particles against heating by the
electron beam during examination, especially at high magnifications. It is also
advantageous that in case of any evaporation from the aerosol particle the metal
replica of the aerosol particle is still visible.
The aerosol particles were shaded with platinum (Pt) at an angle of $\arctan(0.5) = 26°$
(Okada, 1983) in a vacuum chamber at $10^{-6}$ mbar. Pt was evaporated from a Pt wire
($\varnothing$ 0.2 mm, 20 mm length). The Pt wire was drawn around a tungsten (W) wire and
evaporated clusters of Pt atoms when the W wire was heated up electrically by a
85 mA current for 30 sec. The shading procedure produces a layer of Pt particles of 1-
2 nm in diameter on the TEM grid.
After shadowing the TEM grids were examined in TEM using a JEOL JEM-2100
high-resolution instrument, equipped with a $LaB_6$ filament and a Si/Li detector crystal.
The TEM grid containing the aerosol particles was mounted on a sample holder made
of Beryllium to avoid background signals from the sample holder material in the EDX
measurements (see chapter 2.2.3). A CCD camera (Gatan SC1000 Orius, 11
Megapixel) in bottom mount position was used to image the aerosol particles. Images
were taken at high vacuum, less than $35 \times 10^{-5}$ Pa and at an accelerating voltage of
100 kV.





Particles were imaged on TEM grid squares along a diagonal from the center of the
grid to the edge on 6 to 8 squares. To avoid imaging of particles that were damaged
by prior imaging with SEM an area of the grid was chosen which was not exposed to
any electron beam at high magnifications before. Screening each square for individual
particles was done at a magnification of 30.000, and images were taken at
magnifications between 25.000 and 80.000.
**2.2.3  Elemental X-ray spectroscopy**
The elemental analyses were performed using an energy dispersive X-ray detector
JED-2300 attached to the JEM-2100 TEM. In order to avoid time consuming
realignment of the electron beam and focusing procedures the EDX-analyses were
also performed at an accelerating voltage of 100 kV. The energy range measured was
0-20 keV, counting rate was typically 1053 counts/sec$^{-1}$, life time 30 sec, real time
33.00 sec and dead time 10 %. Generally EDX spectroscopy allows the detection of
elements $\geq$ Be as their photon energies are above 100 eV and thus lie within the X-ray
region of the electromagnetic spectrum (Egerton, 2008). Nevertheless the detection of
light elements like C, N, and O that are typical for organic compounds can be difficult
on a Formvar-coated copper grid since the signal intensity can be biased by
attenuation of the X-ray signal through absorption by the adjacent copper grid. For
this reason these elements were not reliably detected and are thus not part of this
study.
Blank grids shadowed with platinum were used to identify the background noise and
signals from the TEM grid including copper from the grid and the Formvar film and
the Pt shadowing. The EDX spectra of blank grids showed only signals from Pt, the
supporting copper TEM grid as well as carbon and oxygen signals from the Formvar
substrate-film (Fig. S1(F)).
**2.2.4  Digital image analysis**
Images taken with SEM at a magnification of 40.000 were evaluated using an
optimized commercial image processing software (Aphelion™ Dev 4.10). In brief,
the maximal intensity of the neighboring background of each aerosol particle was
determined. Using exactly the same image but including the aerosol particle allowed



the separation of the particle and measurement of the particle area in pixels
(Hamacher-Barth et al., 2013).
The particle size was calculated according to Eq. (1)
$D_{pa} = 2\sqrt{Area/\pi}$          (1)
with $D_{pa}$ as the particle equivalent diameter, which is the diameter of a circle that
comprises the same area as the aerosol particle projected onto a two-dimensional
surface (Allen, 1997; Hinds, 1999). The value for the area is calculated from the
number of pixels counted for each particle. A number size distribution of the aerosol
sample was obtained using MATLAB 2011a and the freely available software
package EasyFit.
**3    Results and discussion**
To verify that a representative fraction of the aerosol population has been captured
with SEM we firstly calculated a number size distribution of all aerosol particles and
compare it with measurements from TDMPS. Secondly we sorted all particles imaged
according to morphological similarities into three gross groups, named: single
particles (SP), gel-like particles (GP) and halo particles (HP), shown in Fig. 2. Thirdly,
to obtain more subtle insights into the morphological features of the collected aerosol
particles and simultaneously assess their elemental composition we investigated a
subpopulation of the aerosol particles with TEM and EDX spectroscopy at very high
resolution.
**3.1    Total number size distributions**
In order to derive an overall number size distribution we imaged in total 3909 aerosol
particles at a magnification of 40,000 with SEM. The number size distribution of all
imaged aerosol particles exhibited a bimodal feature with a maximum in the Aitken
mode region at 32 nm in diameter and a double peak above 70 nm in the accumulation
mode region with maxima at 89 nm and 147 nm with a shoulder to larger diameters at
around 335 nm (see Fig. 3, red line). Hamacher-Barth et al. (2013) used the same
image mapping method as used in this study and determined the error of sizing for
polystyrene latex spheres of several diameter sizes between 20 nm and 900 nm was
determined. The error values are displayed in Fig. 3, red arrows. For the TDMPS





number size distribution we assumed an error in determining the mobility diameter of
5% across the whole measuring interval (Wiedensohler et al., 2012).
The two approaches show an overall good agreement between their number size
distributions with a similar modal structure with an Aitken mode below 80 nm and an
accumulation mode at higher diameters. The reduced particle number concentration in
the Aitken mode seen by SEM was probably caused by their partly weak contrast to
the Formvar film, which either resulted in an underestimation of the size or that the
particles remained undetected. The accumulation mode was separated into a double
peak with particle number maxima at 89 nm and 147 nm in SEM and 106 nm and 158
nm in diameter in TDMPS. The aerosol particles at diameters > 100 nm often showed
a   patchy   and   inhomogeneous   appearance   which   might   have   lead   to   an
underestimation of their size and the observed shift to smaller diameters in SEM, at
147 nm compared to 173 nm in TDMPS and broadening of the maximum at 335 nm.
In general the number size distributions determined for particle sizes larger than 20
nm in diameter showed the typical modal features of aerosol collected in the high
Arctic summer boundary layer  (Covert et al., 1996; Heintzenberg et al., 2006) with
an Aitken mode between 26 nm and 80 nm and the multimodal accumulation size
range between 80 nm and 1000 nm (Heintzenberg and Leck, 2012) with the so called
Hoppel minimum around 80 nm inbetween (Hoppel, 1986).
## 3.2    Single particles
Single particles (SP) seen by SEM appeared as single entities that mostly contrasted
sharply and thus could be easily separated from their Formvar background for
diameters > 40 nm. At smaller diameters, however, the contrast to the backgroud was
often weak and probably resulted in an underestimation of particle size or non-
detection of particles. Imaged examples of SP are shown in Figure 4. Of the 3909
particles that were mapped SP were the overall dominating type of particles, 82% of
the total aerosol particles were attributed to this group (Table 1). They were observed
over the whole size range, between 15 nm and 800 nm in diameter with a broad
Aitken mode peaking at 64 nm accompanied by a less pronounced peak at 27 nm. The
majority of SP (80%) appeared in the Aitken mode size region and below 80 nm in
diameter (Table 1). In the accumulation mode size range 18% of SP appeared between
80 and 200 nm with a maximum at 106 nm in diameter and the remaining 2% of the
SP were detected in diameter sizes (Table 1) > 200 nm (Fig. 5, upper panel). We
observed that 35% of the SP partly evaporated under the SEM electron beam but



retained their outer shape on the timescale of minutes. Also at a higher magnification
using TEM the same behavior was seen for 30% of the particles. We tentatively
assigned these particles to be ammonium (bi)sulfate particles. We were guided by the
results published by Heard and Wiffen (1969) and Bigg and Leck (2001b) where
particles with the same morphological features and instability under the electron
microscope were made up by ammonium sulfate, bisulfate or methane sulfonate
mixtures. The presence of ammonium sulfate or bisulfate particles would be
supported by the fact that ammonia has been reported to be the predominant base in
the remote marine troposphere (Söderlund, 1982) that undergoes primarily acid-base
reactions with non-seasalt $H_2SO_4$, an oxidation product of biogenic dimethyl sulfide,
DMS (Quinn et al., 1987). Leck and Persson confirmed the presence of ammonia
bisulfate particles both along the marginal ice edge and over the inner parts of the
pack ice. Over remote marine locations at lower latitudes Meszaros and Vissy (1974)
observed, by means of electron microscopy, ammonium bisulfate concentrations up to
38%, on average 24%, with the highest particle concentrations between 100 nm and 1
um in diameter. In the literature chemical tests have also been used to identify
ammonium and sulfate in samples investigated by TEM (Bigg and Leck, 2001b).
Such tests were not implemented during this study since the use of chemicals would
have added additional mass onto the particles. This would have altered the size and
the morphology of the particles and hampered the investigation of the aerosol
particles with TEM and EDX spectroscopy.
The presence of biogenic nitrate as a counter ion to ammonium can be considered
rather unlikely since the formation of ammonium nitrate happens only after all sulfate
has been neutralized (Kuhn et al., 2010). Nitrate concentrations from impactor
measurements during ASCOS show nitrate values that are one order of magnitude
lower than sulfate concentrations at the same time, usually below 0.1 nmolm$^{-3}$ (*C.*
*Leck pers. comm., 2015*). Moreover ammonium nitrate does not evaporate and is
stable under the electron beam (Rao et al., 1989). To minimize biases due to
evaporative losses and beam damage the ammonium sulfate particles were imaged as
quickly as possible. Figure 6 (upper panel) shows the number size distribution of the
ammonium sulfate particles derived from the TEM images with a maximum at 172
nm in the accumulation mode.
The remaining 65% of the imaged particles (Fig. 5, upper panel) were stable under the
heat of the electron beam and showed no sign of evaporation or changes in
morphology. Some of those particles appeared as skeletal structures (Fig. 4B) that





collapsed and merged to an unstructured flat appearance after exposure times to the
electron beam significantly longer than the justified time for imaging of the particles.
None of the SP particles showed an apparently crystalline appearance that could be
attributed to sea salt or any other inorganic crystalline matter.
### 3.3 Gel-like particles
Aerosol particles classified as gel-like particles (GP) using SEM showed an
amorphous texture with an inhomogeneous distribution of pixel intensity. Their
diffuse structure and weak contrast to the Formvar-film suggested that these particles
predominantly contain light elements like C, H, N and O, which are typical
components of organic matter. The contrast between the particles and the Formvar-
film provides (indirect) information about the elemental composition of the aerosol
particle since the number of the detected secondary electrons increases with
increasing atomic number of the elements present in the aerosol particle (Zhou et al.,
2006) suggesting that the aerosol particles under investigation are built up by matter
of biological origin. The potential similarity in chemical composition between the GP
and the Formvar-film might have lead to an underestimation of the particle size which
resulted in the shift of the higher accumulation mode peak at 173 nm in TDMPS to
147 nm in the total number size distribution (Fig. 3).
GP appeared in the Aitken mode at diameters above 45 nm but were most frequently
observed in the accumulation mode with a maximum at 174 nm, covering all sizes up
to 800 nm (Fig. 5, middle panel). In total 11% of the 3909 particles that were imaged
were classified as GP of which 24% were observed in the Aitken mode > 45 nm, 49%
appeared in the accumulation mode between 80 nm and 200 nm and 27% were
observed > 200 nm (Table 1).
Particles classified as GP were further evaluated with TEM. The higher resolution of
the TEM images revealed better insights into the morphology of the particles and the
GP could be further divided into subgroups (see Fig. 2). 14% of the particles consisted
of a conglomeration of smaller spherical subunits that were welded together and
formed small chains or agglomerates (Fig. 7A, B). Those particles were named
*"aggregate"* particles. 29% of the gel particles appeared as *"aggregate with film"*
particles where *"aggregate"* particles were covered with a diffuse and nearly
electron-transparent film that partly obscured the underlying subunits and produced a



more smooth appearance compared to the bare *"aggregate"* particles (Fig. 7 C, D and E). However, the majority of GP, 57%, showed a *"mucus-like"* texture that was many times widely outspread on the Formvar-film (Fig. 8 A, B), partly in long drawn-out structures (Fig. 8 C) or with small electron dense inclusions (Fig. 8 D).

The individual subunits of *"aggregate"* particles and the dense spots in *"mucus-like"* particles exhibit diameters between 11 nm and 109 nm with a maximum in number size distribution at 39 nm and a smaller maximum at 28 nm. Fig. 9 compares the size distributions of *"aggregate"* components from this study (red line) with those from previous studies in the high Arctic and at lower latitudes (Leck and Bigg, 2005a; 2008; 2010). Similarity with previous studies outside and within the pack ice (Leck and Bigg, 2005b; Orellana et al., 2011, same period as this study) strongly suggests the presence of airborne marine gels. Entanglements, ionic or hydrophobic interactions and/or hydrogen bonds stabilize the three-dimensional biopolymer networks of the marine polymer gels, with electrostatic bonds being the most dominating (Orellana and Leck, 2015). Crosslinking by $Ca^{2+}$ or $Mg^{2+}$ ions seems to be the dominating ionic interaction (Verdugo, 2012). Free biopolymer chains (macromolecules) first assemble into nanogels (100 - 200 nm) that can further anneal into microgels (> 1000 nm) by interpenetration and entanglement of neighboring nanogels or hydrophobic interaction.

Embedded in the polymer network is the high content of water (99%) that prevents the network from collapsing (Chin et a., 1998). The biopolymer networks of marine gels are water solvable, show refractory properties and are highly surface-active. The marine polymer gels are therefore not expected to evaporate under the electron beam.

### 3.4 Halo particles

Besides SP and GP particles were observed that showed a halo-like appearance on the TEM grid with a relatively large central particle surrounded by numerous smaller satellite particles (for examples see Fig. 10). Generally the number of halo particles was however relatively low, on average 7% of the total number of aerosol particles (Table 1).

We will assume that the HP originally existed as one particle in the atmosphere because of the regular structure of the satellite ring around the central particle that through impaction onto the TEM grid splashed out into halo-like particles (HP). The



HP appeared at diameters larger than 75 nm and thus mainly in the accumulation
mode with number maxima at 161 nm and 293 nm, respectively (Fig. 5, lower panel,
left). The very weak contrast of the satellite particles against the Formvar background
(Fig. 3) probable shifted the particle number size distribution towards smaller sizes to
some extent.
Imaging with TEM allowed a more detailed investigation of the HP and revealed
three morphologically different types of the central particle. Two of the particle types
consisted predominantly of particles with skeletal structures and of particles in the
form of *"aggregates"* or *"aggregate with film"*. The third group consisted of particles
that were in opposite to the former particle types unstable under the electron beam
(Fig. 2). Central particles of skeletal structures or *"aggregate"/"aggregate with film"*
made up for 19% and 22%, respectively of the HP examined. Examples of both
particle types are shown in Figs. 10(A) and 10(B). The majority of central particles
(59%), however seemed to partly evaporate during the imaging process, leaving more
transparent structures behind, similar to the SP described in chapter 3.2. We sized the
central particles individually in order to compare them with particles of similar
morphology to the SP or GP. The number size distribution of the *"aggregate"*,
*"aggregate with film"* and skeletal particles is shown in Fig. 6(B), green line, with a
maximum at 270 nm diameters, compared to the maximum at 171 nm (Fig. 6(B), red
line) resulting from sizing the heat sensitive central particles under the electron beam.
The satellites (particulates or droplets) exhibited varying morphologies. Sometimes
numerous small satellites surrounded the central particle in a symmetrical ring (Fig.
10(A) whereas in other cases the satellite droplets were larger but fewer (Fig. 10(B),
(C)). In the literature three types of compounds have been described to form satellites
when airborne aerosol particles impact on a collection substrate: sulfuric acid (Ayers,
1978; Ferek et al., 1983), ammonium sulfate and bisulfate (Bigg, 1980; Busek and
Pósfai, 1999) and methane sulfonic acid, MSA, (Bigg et al., 1974). Sulfuric acid
exhibits a distinctive morphology: a central particle surrounded by a droplet-halo of
numerous smaller satellites. Neutralization of sulfuric acid by ammonium or a high
content of methane sulfonic acid produces a halo of larger and fewer droplets (Bigg
and Leck, 2001a). The morphology of the droplet-halos we observed in this study
points towards the presence of sulfuric acid, often in a mixture with ammonium
sulfate or bisulfate and/or methane sulfonic acid. As discussed in section 3.2 these
sulfur-containing components have not only been reported to be present over the



Arctic pack ice area in summer (Bigg and Leck, 2001a) but also to frequently occur in
the remote marine atmosphere (Barnard et al., 1994; Capaldo and Pandis, 1997; Kettle
et al., 1999). The observed number size distribution for all HP (Fig. 5, lower panel) is
in agreement with results from the high Arctic reported by Hillamo et al. (2001)
which observed the first maximum in sulfate containing aerosol particles at diameters
> 80 nm and in ammonium and MSA containing particles at diameters > 100 nm.
**3.5    EDX measurements**
To determine the elemental composition of the aerosol samples an EDX spectrometer
coupled to TEM was used. EDX spectra of 103 aerosol particles were recorded in
conjunction with the imagining process. Molecular dynamics studies on
polysaccharides by Li et al. (2013) and Sun et al. (2014) have shown that not only the
divalent ions $Ca^{2+}$ and $Mg^{2+}$ but also the monovalent ions $Na^+$ and $K^+$ can stabilize the
three-dimensional biopolymer gel networks. Inspired by these results we focused on
the detection of the alkali ions $Na^+$ and $K^+$ and the divalent ions $Ca^{2+}$ and $Mg^{2+}$ in the
gel-type particles. In the following we will refer to $Na^+$ and $K^+$ as $Na^+/K^+$ and $Ca^{2+}$
and $Mg^{2+}$ as $Ca^{2+}/Mg^{2+}$.
The analysis revealed the following characteristics: $Na^+/K^+$ was detected in 91% of
the SP, 13% of these particles contained exclusively $Na^+/K^+$ whereas 78% contained
both types of metal ions, $Na^+/K^+$ and $Ca^{2+}/Mg^{2+}$ with the latter only in minor
quantities (Fig. 11(A)); examples of EDX spectra for the different types of particles
are shown in Fig. S1. The *"aggregate"* particles contained exclusively $Na^+/K^+$ in 20%
of the particles and predominantly $Na^+/K^+$ and minor contents of $Ca^{2+}/Mg^{2+}$ in 80% of
the particles (Fig. 11(B)). For the particle types *"aggregate plus film"* and *"mucus-*
*like"* particles, however, a clear dominance of $Ca^{2+}/Mg^{2+}$ was detected. 17% of the
*"aggregate plus film"* particles contained only $Ca^{2+}/Mg^{2+}$ and 67% of the particles
contained $Ca^{2+}/Mg^{2+}$ accompanied by minor contents of $Na^+/K^+$ (Fig. 11(C)).
*"Aggregate plus film"* particles thus contained to 86% $Ca^{2+}/Mg^{2+}$ as the dominating
type of ions. *"Mucus-like"* particles contained to 11% only $Ca^{2+}/Mg^{2+}$ and up to 86%
$Ca^{2+}/Mg^{2+}$ accompanied by minor contents of $Na^+/K^+$. 97% of the type *"mucus-like"*
particles thus contained $Ca^{2+}/Mg^{2+}$ as the dominating type of metals. In summary, we
observed a gradual transition from particles with a dominating content of $Na^+/K^+$ to
particles with a dominating content of $Ca^{2+}/Mg^{2+}$ moving from SP over *"aggregate"*
particles and *"aggregate with film"* particles to *"mucus-like"* particles. We therefore





hypothesise a connection between the morphology of the particles and the respective
dominating crosslinking ion within the polymer 3-dimensional network of the marine
gels. Li et al. (2013) compared polysaccharides with 3 and 4 repetition units of
molecular weights of 1.9 kDa and 2.5 kDa respectively as representations for organic
matter in seawater (Verdugo, 2004). Their results showed that the assembly of the
longer polysaccharide chains seems to be accelerated in the presence of $Ca^{2+}$ whereas
the presence of $Na^+$ has a positive effect on the assembly of shorter polysaccharide
chains. Considering the observed morphology of the aerosol particles that are built up
by marine gel matter a high content of $Ca^{2+}/Mg^{2+}$ could facilitate the formation of
fluffy and less compact *"mucus-like"* gel matter whereas the presence of $Na^+/K^+$
favoured a more compact structure of type *"aggregate"* and SP.
Halo particles with a center of gel or fraction of a gel showed a high content of
$Na^+/K^+$: 50% of the particles contained those metals exclusively whereas another 25%
contained mainly $Na^+/K^+$ with smaller amounts of $Ca^{2+}/Mg^{2+}$ (Fig. 12). The high
content of alkali metal ions in those particles suggested that they originated from SP
or fragmented *"aggregate"* particles, which were exposed to processes that lead to
condensational growth of the original gel particles.
**4   Summary and conclusions**
Aerosol particles collected in the summer high Arctic north of 80° were individually
and objectively investigated with electron microscopy and subsequent image mapping.
This enabled a division of the aerosol particles into three size resolved gross
morphological groups, single particles (SP), gel particles (GP) and halo particles (HP).
Single particles (SP) dominated the aerosol population in terms of numbers and made
up 82% of all particles; they were observed over the whole sub-micrometer size range
and clearly dominated the Aitken mode. The majority of SP (65%) was stable under
the electron microscope and showed no signs of evaporation or morphological
changes during imaging. These particles with refractory properties appeared over the
whole size range of particles observed whereas the remaining 35% of SP appeared to
be heat instable, evaporated partly and were predominantly observed in the
accumulation mode. GP were observed at diameters > 45 nm predominantly in the
accumulation mode with a maximum in number at 154 nm and contributed with 11%
to the total particle number. The GP exhibited various morphological features and



appeared as *"aggregate"* particles (14%) and as *"aggregate with film"* particles
(29%) but the majority, 59% was made up by *"mucus-like"* particles. 70% of the GP
in our study showed to be smaller than 100 nm in diameter, and measured 90%
smaller than 200 nm.
HP appeared mainly in the accumulation mode at diameters > 60 nm with a maximum
in number at 161 nm and contributed up to 7% to the total particle number mapped.
The majority of HP consisted of heat instable particles, probably ammonium bisulfate
(59% of all particles), internally mixed with sulfur containing compounds (sulfuric
acid, ammonium bisulfate, methane sulfonic acid). The remaining fraction was made
up by *"aggregate"* particles (19%) and decomposed or fragmented gel matter (22%)
internally mixed with sulfur containing compounds.
Electron dispersive X-ray (EDX) spectroscopy revealed a gradual transition in the
content of $Na^+/K$ and $Ca^{2+}/Mg^{2+}$ between different particle morphologies. SP and
*"aggregate"* particles preferentially contained $Na^+/K^+$ whereas *"aggregate with film"*
particles and *"mucus-like"* particles contained mainly $Ca^{2+}/Mg^{2+}$ with minor contents
of $Na^+/K^+$. Supported by model studies (Li et al., 2013; Sun et al., 2014) we
hypothesize that a correlation exists between particle morphology and the prevalence
of the ions $Na^+/K^+$ and/or $Ca^{2+}/Mg^{2+}$ were the prevalence of $Ca^{2+}/Mg^{2+}$ facilitates the
formation of large organic assemblies of GP type whereas a lack of $Ca^{2+}/Mg^{2+}$ and a
prevalence of $Na^+/K^+$ prohibit the formation of large assemblies leading to smaller
entities of SP type.
The so far generally insufficient understanding of the size resolved aerosol
composition and especially the role of organic compounds, their morphology and state
of mixture had hampered a detailed understanding of the of the processes that lead to
the activation of the high Arctic aerosol particles and thus their role in the formation
of cloud droplets (Leck and Svensson, 2015; Martin et al., 2011; Zhou et al., 2001;
Leck et al., 2002). One evident outcome from this study is that the aerosol particles to
be activated into cloud droplets over the Arctic pack ice areas cannot be seen as
simply inorganic salts. None of the aerosol particles showed an apparently cubic or
otherwise crystalline appearance that could be attributed to sea salt particles. Instead
the results from this study clearly show that organic marine gel matter significantly
contributes to the particle number concentration over the whole sub-micrometer size
range but especially at diameters below 60 nm.





A parallel study conducted during the ASCOS campaign (Orellana et al., 2011)
demonstrated that airborne aerosol particles contain hydrophobic moieties on their
surface which play an important role for gel formation (Maitra et al., 2001) and
increase the rate of gel assembly (Ding et al., 2008). The interaction of the
hydrophilic and hydrophobic entities on the structure of the polymer gels likely will
influence the water vapor pressure and decrease the surface tension of the cloud
droplets to be formed (Leck and Svensson, 2015; Ovadnevaite et al., 2011).
Water-soluble particles like ammonium sulfate were present mainly in the
accumulation mode at diameters above 100 nm. Growth of the sub-Aitken particles
probably resulted from deposition of acids/organic vapors on polymer gel particles
and produced HP or sulfur containing particles with hygroscopic properties typical for
a gel nucleus covered by a sulfate-methane sulfonate-ammonium complex. At the
same time the fragmentation of larger particles is capable of adding numbers into the
Aitken mode (Leck and Bigg, 1999; 2010; Karl et al., 2013). Orellana and Verdugo
(2003) and Orellana et al. (2011) observed the sensitivity of marine gels to changes in
the physicochemical environment (pH and T) and the fragmentation of gel matter into
smaller entities as a result to UV radiation exposure.  Acidic compounds typically
found in the marine atmosphere like sulfuric acid and dimethyl sulfide (DMS) have
induce volume collapse of the swollen hydrated polymer gel network into a
condensed and more compact form (Tanaka et al., 1980; Orellana et al., 2011).
Condensation of sulfur acidic compounds and in-cloud processing of the marine gels
in the atmosphere during their passage over the pack-ice and continuous exposure to
UV radiation due to the length of the polar day in summer could produce smaller
sized fragments of marine gels similar to the spherical subunits observed in
*"aggregate"/"aggregate with film"* particles and the dense inclusions in *"mucus-like"*
particles. Since SP showed a maximum in number concentration in the same size
range, at 27 nm, it cannot be excluded that fragmentation of gel matter or pH induced
collapse of the gels lead to the formation of smaller entities and by that providing a
mechanism to produce SP and to add particle numbers to the Aitken mode.
In hope of enhancing our understanding on CCN properties promoting /suppressing
cloud droplet formation over the pack ice area in summer and at the same time meet
the demand for observational data for the evaluation of climate models, this study has
presented critical size resolved data on particle morphology, chemical composition
and state of mixture based on the analysis of individual particles.





**Acknowledgements**
This work is part of the ASCOS (Arctic Summer Cloud Ocean Study). ASCOS was
an IPY project under the AICIA-IPY umbrella and an endorsed SOLAS project.
ASCOS was made possible by funding from the Knut and Alice Wallenberg
Foundation, the DAMOCLES Integrated Research Project from the European Union
6th Framework Program and the Swedish National Research Council (VR). The
Swedish Polar Research Secretariat provided access to the icebreaker Oden and
logistical support. The authors thank A. Held for collecting the aerosol particles
during ASCOS, and A. Öhrström and C. Rauschenberg for their help with sample
imaging / EDX measurements.



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



|  | 15-80 nm | 80-200 nm | > 200 nm | Sum (% of total) |
|---|---|---|---|---|
| Single particles (% of total SP) | 2609 (80%) | 573 (18%) | 57 (2%) | 3239 (82 %) |
| Gel particles (% of total GP) | 97 (24%) | 198 (49%) | 108 (27%) | 403 (11 %) |
| Halo particles (% of total HP) | 9 (3%) | 95 (36%) | 163 (61%) | 267 (7 %) |
| Total number of particles | | | | 3909 (100%) |

2  Table 1. Numbers and percentage of total for single particles (SP), gel particles (GP)

3  and halo particles (HP) imaged with SEM and used for calculating number size

4  distributions.



| | No $Na^+/K^+$, no $Ca^{2+}/Mg^{2+}$ | $Ca^{2+}/Mg^{2+}$ | $Na^+/K^+$ | $Ca^{2+}/Mg^{2+}$ and $Na^+/K^+$ |
|---|---|---|---|---|
| Single particles | 4 % | 4 % | 13 % | 78 % ($Na^+/K^+$) |
| *"Aggregate"* particles | - | - | 20 % | 80 % ($Na^+/K^+$) |
| *"Aggregate with film"* particles | - | 17 % | 17 % | 67% ($Ca^{2+}/Mg^{2+}$) |
| *"Mucus-like"* particles | - | 11 % | 3 % | 86 % ($Ca^{2+}/Mg^{2+}$) |
| Halo particles | 25 % | - | 50 % | 25 % ($Na^+/K^+$) |

Table 2. Fraction of particles containing the ions $Na^+/K^+$ or $Ca^{2+}/Mg^{2+}$, or both, $Na^+/K^+$ and $Ca^{2+}/Mg^{2+}$, or neither Na/K nor $Ca^{2+}/Mg^{2+}$ in single particles, *"aggregate"* particles, *"aggregate with film"* particles, *"mucus-like"* particles and halo particles. The ions written in brackets in the last column indicate the prevalent type of ion in the respective type of particle.




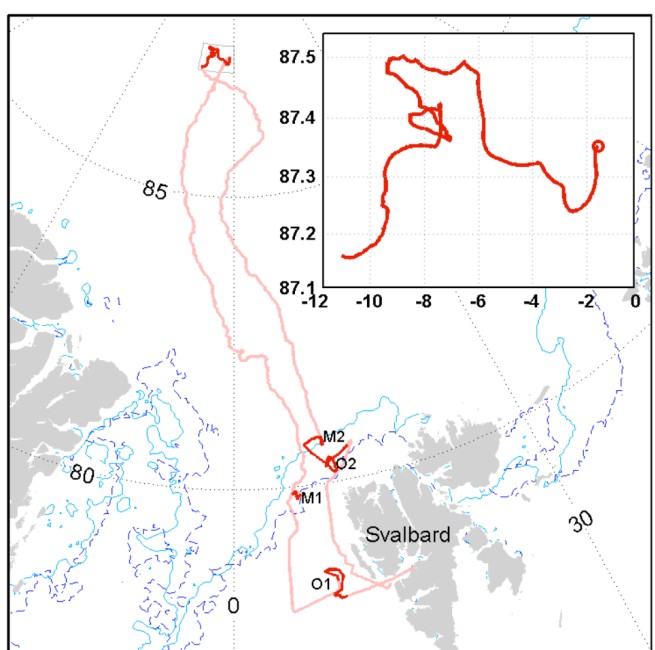

Figure 1. Track of the icebreaker *Oden* in the Arctic (pink). The path during the ice-drift is shown in the insert (red line); the circle indicates the start of the ice-drift, the ice edge (thin blue line) was passed on 12 August 2008.





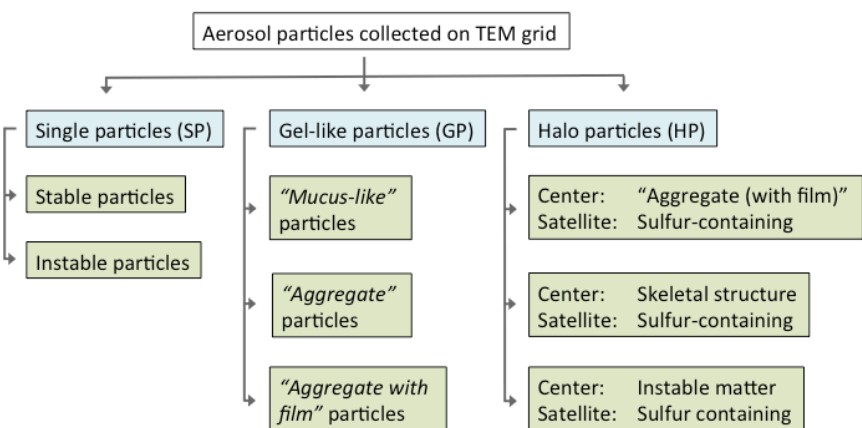

Figure 2. Scheme of the aerosol particle types collected on Formvar grid; the particles observed with SEM are shaded in light blue, particles observed in TEM are shaded in light green.





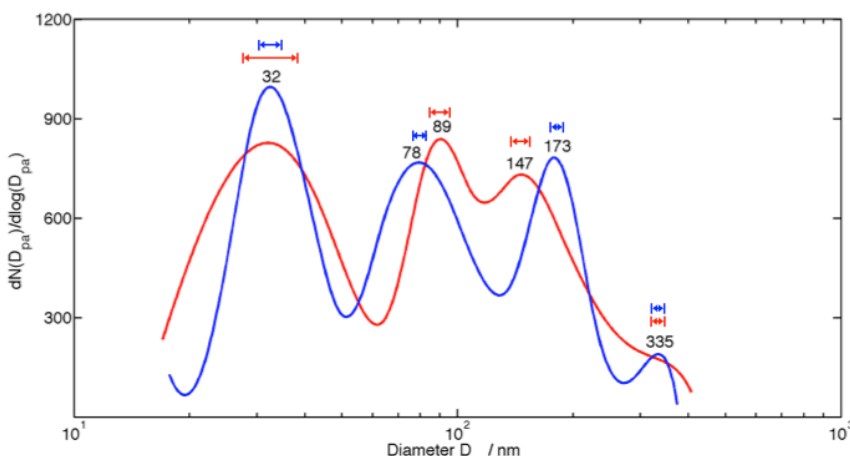

Figure 3. Number size distribution of the total aerosol collected for this study; red line: SEM derived particle number distribution, error bars represent the error size determination retrieved from calibration measurements described in Hamacher-Barth et al. (2013); blue line: number size distribution from simultaneous TDMPS measurements, errors bars represent 5 % uncertainty of the data (Wiedensohler et al., 2012).

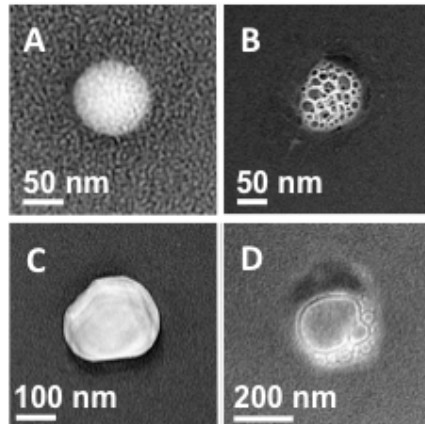

Figure 4. Examples for single particles (SP) observed with TEM (A) a particle stable under the beam of the electron microscope (B) particle with a skeletal structure (C) a particle stable under the electron beam (D) an example for a particle that is unstable under the electron beam.



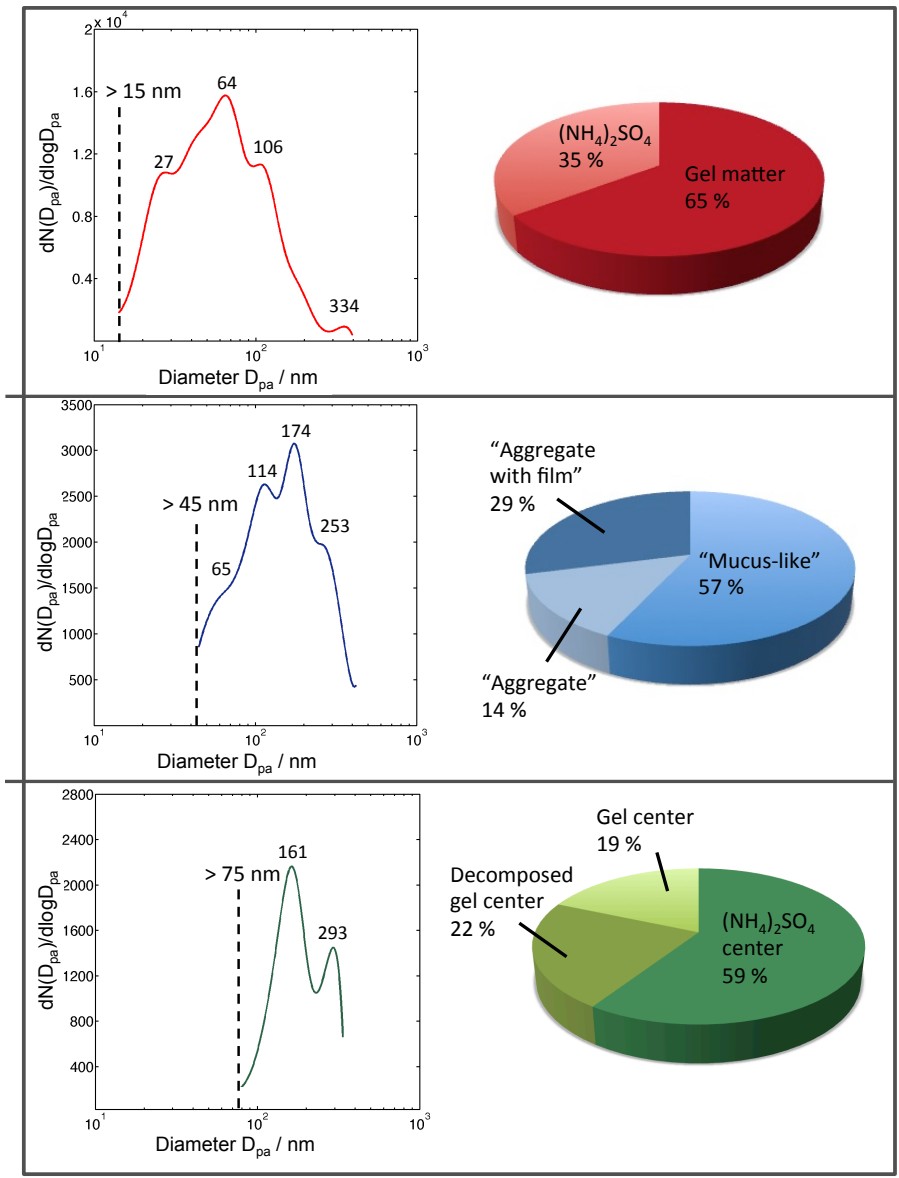

Figure 5. SEM Size distribution of the particle types evaluated for this study (to the left) plus the relative contributions of the different subgroups of particles derived from TEM (to the right); the dashed line in each figure marks the lowest diameter in which the respective particle type appears. Upper panel: single particles, middle panel: gel particles; lower panel: halo particles.



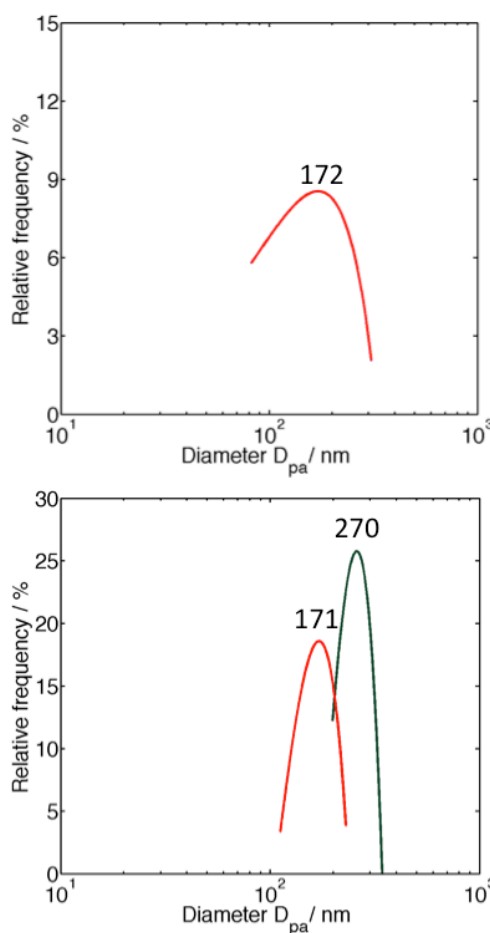

Figure 6. Number size distributions obtained from TEM images. Upper panel: instable SP; lower panel: central particles of HP; red line: instable central particles, green line: *"aggregate"*, *"aggregate with film"* and skeletal particles.



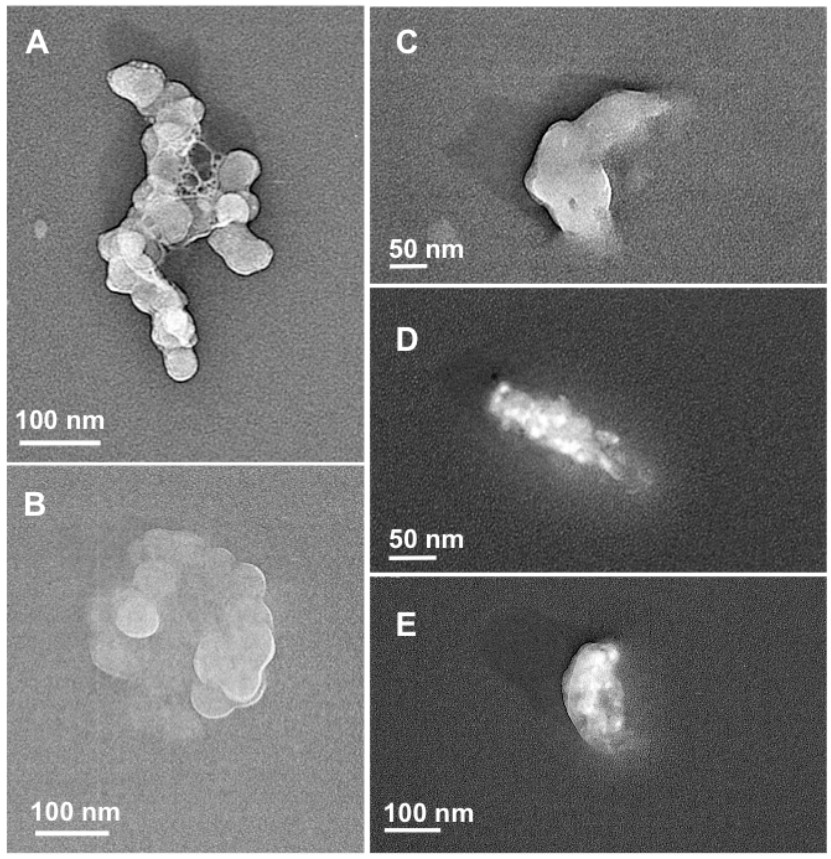

Figure 7. Examples for particles built up by aggregated subunits, observed with TEM; (A) and (B) consist of pure aggregates (*"aggregate"* particles); (C), (D) and (E) aggregate particles covered with a thin film of gel *("aggregate with film"* particles).



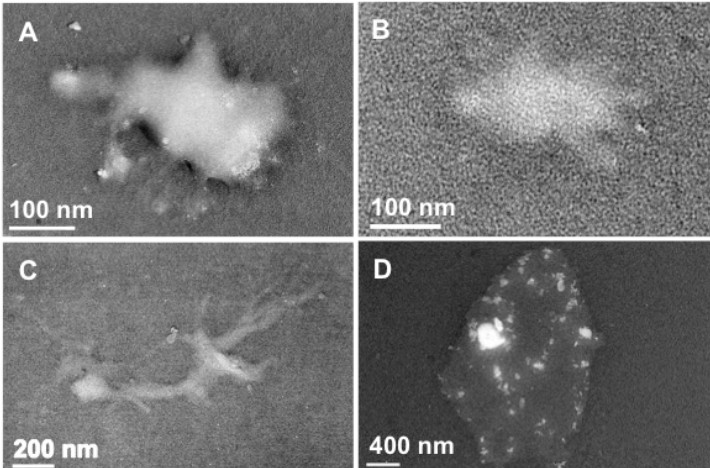

Figure 8. Examples for *"mucus-like"* particles observed with TEM. (A) mucus matter with small dense inclusions, partly outdrawn on the Formvar film, (B) mucus matter, outdrawn on the Formvar film, (C) mucus matter, extensively outdrawn on the formvar film, (D) mucus matter with numerous dense inclusions





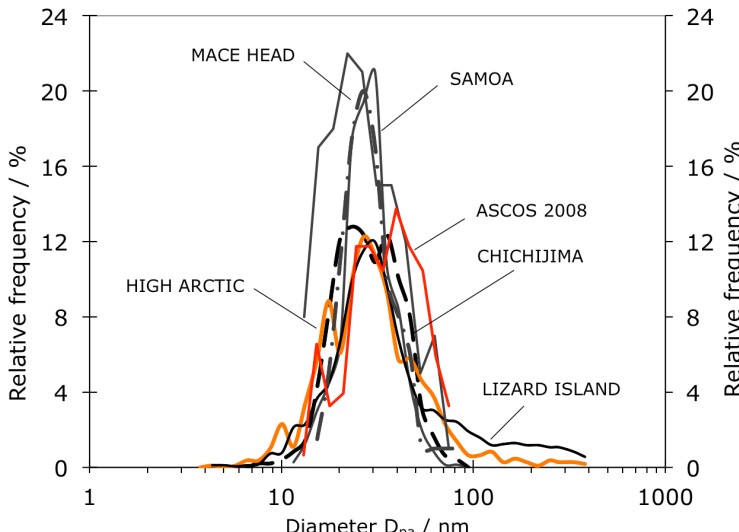

Figure 9. Number size distributions of airborne aggregate particles and their building blocks at different locations: Mace Head (53°N, 10°W), Lizard Island (14.6°S, 145.5 °E), American Samoa (14 °S, 172 °W), Chichijima (27 °N, 142 °E), High Arctic (AOE-2001, between 88.9 °N and 88.2 °N; orange line), and ASCOS 2008 (between 87 °N, 1°W and 87 °N, 11 °W, red line). All particles were assumed to be spherical in shape (from Bigg and Leck (2008), modified).


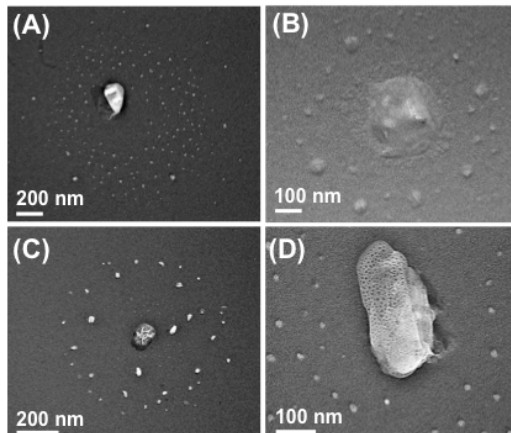

Figure 10. Examples for Halo Particles observed with TEM; (A) a central *"aggregate"* particle surrounded by satellite particles of sulfuric acid and a smaller amount of methane sulfonic acid; (B) central *"aggregate with film"* particles, surrounded by satellite particles of sulfuric acid and methane sulfonic acid; (C) central particle formed by ammonium sulfate, satellite particles formed by methane sulfonic acid, probably mixed with sulfuric acid. (D) central particle of degenerated gel, surrounded by methane sulfonic acid mixed with sulfuric acid.



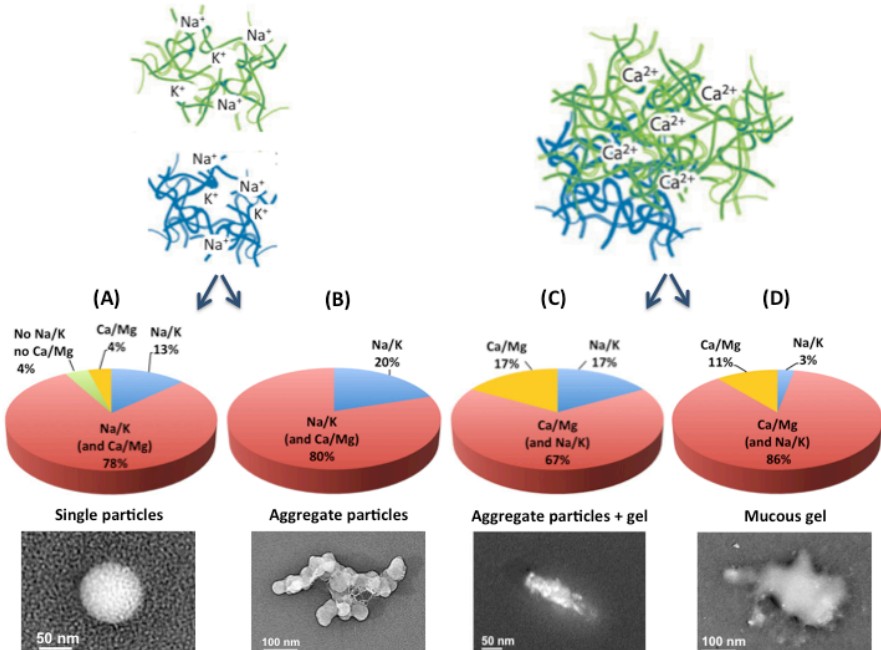

Figure 11. Fraction of particles containing the following ions: Na/K (blue), Ca/Mg (yellow), Na/K and Ca/Mg (red), and neither Na/K nor Ca/Mg (green); (A) Single particles comprised of gel matter, (B) *"Mucus-like"* particles, (C) *"Aggregate"* particles, (D) *"Aggregate with film"* particles.





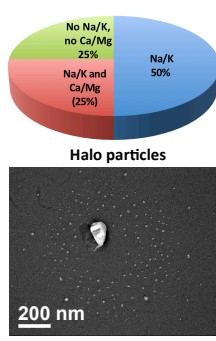

Figure 12. Fraction of HP containing the following ions: $Na^+/K^+$ (blue), $Na^+/K^+$ and $Ca^{2+}/Mg^{2+}$ (red), and neither $Na^+/K^+$ nor $Ca^{2+}/Mg^{2+}$ (green).