# Peer review of "Size resolved morphological properties of the high Arctic summer aerosol during ASCOS-2008"

_Atmospheric Chemistry and Physics, 2015_

## Referee Comment (RC1) · Anonymous Referee #2 · 12 Feb 2016

The paper presents a hard-earned data set of single particle analysis of samples collected in the high Arctic during the summer of 2008. The analysis is used to categorize particles according to their morphological and chemical properties. The most significant result is that "the particles to be activated into cloud droplets over the Arctic pack ice areas can not be seen as simply organic salts". This result has implications for the modeling of CCN activity and cloud drop formation over the high Arctic. The paper should be published once the concerns outlined below are addressed.

p. 2, line 14: I'm not sure how particles can increase planetary albedo by absorbing sunlight.

p. 4, line 5: Is this meant to be Arctic SURFACE OCEAN DOM?

p. 5, line 21: Should be SVALBARD.

p. 11, line 25: Why is the size distribution characterized as "bimodal" when, on average, it contains 3+ peaks?

Figure 6 caption: I think it should be "unstable", not instable. Also the labeling of the two lines is confusing. In the lower panel there are central particles of HP (red line) and unstable central particles (green line). What line represents the aggregate, aggregate with film and skeletal particles? Also – in this figure and throughout, it would help the reader immensely if legends were added to the plots.

Figure 9: It is very difficult to tell the orange and red line apart.

p. 15, lines 13 – 20: It is unclear if the descriptions of bonds holding the marine polymer gels together are referring to gels in atmospheric aerosol or in seawater. Assuming seawater (given the Verdugo reference), the size range of nanogels (100 – 200 nm) and microgels (> 1000 nm) are larger than the airbone aggregate particles shown in Figure 9. How can this difference be reconciled?

p. 17, line 10: Should be IMAGING.

p. 17, lines 30 – 33: The dominant particle type was those containing both Na/K and Ca/Mg. Stating a "dominating content of Na/K" or Ca/Mg is not accurate.

Figures 5, 6, and 9: Each of these figures contains some version of the aggregate particle type but each shows different size ranges for that particle type. What is the definition of "airborne aggregate particle" shown in Figure 9 (which resides in the 10 to 100 nm size range) compared to the aggregate particle type shown in Figure 9 (which is in a larger size range)? How is this reconciled with the statement on p. 19, lines 30 – 33 that says "organic marine gel matter contributes to the particle number concentration . . ..especially at diameters below 60 nm"?

p. 20, lines 12 – 29: The discussion of the fragmentation of larger particles into smaller particles in the atmosphere due to UV radiation exposure is highly speculative and

not supported by direct evidence. The papers cited appear to be based on studies of seawater. What thermodynamically viable mechanism can break apart particles in the 100 - 200 nm size range in the atmosphere? As far as I know, there are no reported observations of such events. (By the way, the Karl et al. (2013) and Tanaka et al. (1980) references, which may provide some insight here, are missing in the list of citations.)

---

## Referee Comment (RC2) · Anonymous Referee #1 · 23 Mar 2016

It is relatively easy to obtain detailed size distributions and bulk chemistry of aerosols but in order to understand the sources of particles and their individual chemistry, imaging and analysis of separate particles is required. This has not been a popular form of research because it requires the use of electron microscopes and a very large amount of microscope time to study a representative sample of the aerosol. Interpreting the results also requires experience and wide background knowledge of possible sources of the aerosol.

The paper under review is very valuable because it greatly extends previous work of this type on the summer high-Arctic aerosol using improved techniques and studying a remarkably large number of individual particles. I strongly recommend it for publication,

but contribute a few comments that might be useful.

1. Sizing of the aerosol from electron microscope imagery is notoriously difficult and in this case has resulted in acceptable agreement with TDMPS size distributions. Subsequent investigators will want to try to compare their work with that under review. It should therefore be made clear what assumptions were made in sizing particles such as A in figure 4 where no shadow is visible, or B in that figure which shows a thin shadow on the lower edge suggesting that it is a flat crystal. Similarly, in figure 7, were the components of the chain aggregates assumed to be spherical, their volumes summed and the diameter of a sphere with the equivalent diameter calculated? If so, attention should be called to the paper by Rogak et al. (Aerosol Sci. Tech 18, 25-47, 1993) which showed that a mobility analyser bases diameter of such particles on the projected area rather than on the volume. This will affect the comparison of mobility and EM size distributions.

2. P.10 line 24. I can't understand why C and O were not detected on blank films of polyvinyl formal. In the supplemental data the carbon signature is strong, so the detector was sufficiently sensitive.

3. Droplet haloes: I don't believe the splash hypothesis is appropriate for low velocity electrostatic collection. In fact I think it is also doubtful for particles of the size of those in figure 10 collected by high velocity impaction. Stratospheric aerosol sampling by an Ames Research Center group (Farlow and Ferry) 40 years ago found that the sulfuric acid particles did not develop haloes if all contact with water vapour was avoided before examination. (Possibly in JGR 82, 4921-4929, 1977 but I don't have the article) It was later confirmed by laboratory experiments (Bigg, Tellus, 38B, 62-66, 1986). A possible explanation is that acid vapour extends outwards from the captured particle as a monolayer (or multiple layers) on the surface. On exposure to water vapour the molecules take up water and coalesce to form tiny droplets.

4. P.15, line 23: "the biopolymer networks of marine gels are water solvable". Solvable

means that an answer is available for a problem. If you meant "soluble", how could they exist as entities in the ocean?

The manuscript is well-written, the diagrams informative and the references very comprehensive. There are some instances where the spelling or wording differs slightly from conventional English usage and some of these are listed below together with suggested alternatives.

p.5, line 21: Longyearbyen, Svalbard

p.5, line 25 and beyond: Since you are reporting completed work it would be more conventional to use the past tense rather than the future tense. E.g., change "will use" to "used".

p.6, line 2: according to morphological...

p.8, line 19: In order to compare (to) the number...

p.9, line 21: Although wolfram is more logical in view of its symbol, tungsten is the common English usage.

p.10, line 19: were not reliably detected

p.16 line 10: unstable

p.16 line 17: morphology to the

p.20 line 13: is capable of adding

p.20 line 30: In the hope of enhancing

References, p.23, line 5: Ayers, G.P.

p.23 line 8: pouchetii

p.25 line 4: Cambridge

p.30 line 21- 24: Remove hyphens in Ramaswamy, Isaksen, climate and Intergovernmental

p.31, line 6: atmosphere

Figure 10 caption, line 6: degenerated
* * *

---

## Author Comment (AC1) · 11 Apr 2016

**We are grateful to the reviewer for her/his positive comments and careful reading of the manuscript. Below we address the comments with our answers in blue. The numbering of pages and lines in our answers refer to the new version of the manuscript. Changes in the manuscript are written in red.**

**Anonymous Referee #2**

The paper presents a hard-earned data set of single particle analysis of samples col- lected in the high Arctic during the summer of 2008. The analysis is used to categorize particles according to their morphological and chemical properties. The most signifi- cant result is that "the particles to be activated into cloud droplets over the Arctic pack ice areas can not be seen as simply organic salts". This result has implications for the modeling of CCN activity and cloud drop formation over the high Arctic. The paper should be published once the concerns outlined below are addressed.

p. 2, line 14: I'm not sure how particles can increase planetary albedo by absorbing sunlight.
We rephrased the sentence to
P2L13. "They alter the planetary albedo both directly by absorbing and scattering sunlight and indirectly …"

p. 4, line 5: Is this meant to be Arctic SURFACE OCEAN DOM?
Corrected

p. 5, line 21: Should be SVALBARD.
Corrected

p. 11, line 25: Why is the size distribution characterized as "bimodal" when, on average, it contains 3+ peaks?
P11, line 24-28. Thank you for pointing out this unclarity. The sentence was rephrased to:
"The number size distribution of all imaged aerosol particles exhibited a maximum in the Aitken mode region at 32 nm in diameter and a double peak above 70 nm in the accumulation mode region with maxima at 89 nm and 147 nm with a shoulder to larger diameters at around 335 nm (see Fig. 3, red line)."

Figure 6 caption: I think it should be "unstable", not instable. Also the labeling of the two lines is confusing. In the lower panel there are central particles of HP (red line) and unstable central particles (green line). What line represents the aggregate, aggregate with film and skeletal particles? Also – in this figure and throughout, it would help the reader immensely if legends were added to the plots.
Labels were added to the plot for further clarification.

Figure 9: It is very difficult to tell the orange and red line apart.
The color of the orange line was changed to blue.

p. 15, lines 13 – 20: It is unclear if the descriptions of bonds holding the marine polymer gels together are referring to gels in atmospheric aerosol or in seawater. Assuming seawater (given the Verdugo reference), the size range of nanogels (100 – 200 nm) and microgels (> 1000 nm) are larger than the airbone aggregate particles shown in Figure 9. How can this difference be reconciled?
The paragraph has been rewritten to:
p. 15, lines 16-32: "In seawater the observed size range of gel particles ranges from sovated nanogels (100-200 nm; Bigg et al., 2004) that can further anneal into microgels (> 1000 nm) by interpenetration and entanglement of neighboring nanogels or hydrophobic interaction. Changes in environmental factors like UV-B radiation (Orellana and Verdugo, 2003) or physico-chemical parameters like pH and temperature (Tanaka et al., 1980) lead to

inhibition/dispersion or volume change of the gel polymer assemblies. The transport from the ocean water into the atmosphere results in an enhanced exposure of the gel particles to solar UV-B radiation. Together with changes in the physico-chemical environment of the gel particles due to e.g. condensation of acidic gases onto the aerosol droplets the transport into the atmosphere thus might lead to fragmentation and/or shrinking of the gel matter and result in a reduced diameter of atmospheric gel particles compared to gel matter in the ocean (Leck and Bigg, 2005b; Orellana et al., 2011). Embedded in the polymer network is the high content of water (99%) that prevents the network from collapsing (Chin et a., 1998). The biopolymer networks of marine gels are highly surface active and show refractory properties are therefore not expected to evaporate under the electron beam."

p. 17, line 10: Should be IMAGING.
Corrected

p. 17, lines 30 – 33: The dominant particle type was those containing both Na/K and Ca/Mg. Stating a "dominating content of Na/K" or Ca/Mg is not accurate.
Thank you for pointing this out, we corrected a mistake in the figure legend.

Figures 5, 6, and 9: Each of these figures contains some version of the aggregate particle type but each shows different size ranges for that particle type. What is the definition of "airborne aggregate particle" shown in Figure 9 (which resides in the 10 to 100 nm size range) compared to the aggregate particle type shown in Figure 9 (which is in a larger size range)?

Fig. 5 (middle panel) shows the number size distribution of **all gel-type particles** observed with SEM. The SEM images do not allow a differentiation into the subgroups of gel-type particles ("aggregate" particles, "aggregate with film" particles etc.). Gel-type particles were observed above 45 nm and up to 800 nm in diameter.

Fig. 6 (lower panel) shows the number size distributions for **central particles of halo particles**, with the green line representing stable central particles (comprising "aggregate" particles, "aggregate with film" particles and skeletal particles) imaged with TEM, and the red line unstable central particles of halo particles.

Fig. 9 shows the number size distributions of **airborne aggregate particles and their building blocks** at different locations as obtained with TEM. Sizing of *"aggregate"* particles in our study revealed the 10 to 100 nm size range for the building blocks of these particles.

Therefore Figures 5, 6 and 9 show number size distributions of different types of particles that cover different size ranges and thus cannot be readily compared.

How is this reconciled with the statement on p. 19, lines 30 – 33 that says "organic marine gel matter contributes to the particle number concentration . . ..especially at diameters below 60 nm"?
It should be "below 75 nm" instead of "below 60 nm".
Organic marine gel matter (in single particles and gel-type particles) contributes mainly to particle numbers below 75 nm because all other types of particles appear at higher diameters: unstable single particles appear mainly in the accumulation mode (Fig. 6, upper panel) and halo particles (with sulphur-containing satellites) appear above 75 nm (Fig. 5, lower panel).

p. 20, lines 12 – 29: The discussion of the fragmentation of larger particles into smaller particles in the atmosphere due to UV radiation exposure is highly speculative and not supported by direct evidence. The papers cited appear to be based on studies of seawater.

What thermodynamically viable mechanism can break apart particles in the 100 - 200 nm size range in the atmosphere? As far as I know, there are no reported observations of such events. (By the way, the Karl et al. (2013) and Tanaka et al. (1980) references, which may provide some insight here, are missing in the list of citations.)

It is correct that UV induced fragmentation of larger gel particles into smaller particles has been studied in seawater, as discussed in chapter 3.3 (p15, line 21-29). According to the current understanding the transport of gel matter from the ocean into the atmosphere occurs through bubble bursting and subsequent formation of small droplets that comprise seawater and e.g. marine gels (see Introduction, p.4, line 13 ff.). In the atmosphere these droplets can be subject to further condensation of water vapor or other substances, coalescence and incloud processing which all potentially change the physico-chemical conditions within the droplet (Tanaka et al., 1980). The latter in combination with an increased exposure to UV-B radiation in the atmosphere makes it in our view reasonable to discuss fragmentation and/or skrinking as a possible mechanism to produce smaller sized fragments of marine gels (Leck and Bigg, 2010; Karl et al., 2013). This argument is further supported by the finding that spherical subunits in "aggregate"/"aggregate with film" particles and the dense inclusions in "mucus-like" particles appear at diameters down to 10 nm not only in the Arctic but also at different locations at lower latitudes (Fig. 9).

Tanaka et al. (1980) and Karl et al. (2013) were added to the reference list.

---

## Author Comment (AC2) · 11 Apr 2016

**We are grateful to the reviewer for her/his positive comments and careful reading of the manuscript. Below we address the comments with our answers in blue. The numbering of pages and lines in our answers refer to the new version of the manuscript. Changes in the manuscript are written in red.**

**Anonymous Referee #1**

It is relatively easy to obtain detailed size distributions and bulk chemistry of aerosols but in order to understand the sources of particles and their individual chemistry, imaging and analysis of separate particles is required. This has not been a popular form of research because it requires the use of electron microscopes and a very large amount of microscope time to study a representative sample of the aerosol. Interpreting the results also requires experience and wide background knowledge of possible sources of the aerosol.

The paper under review is very valuable because it greatly extends previous work of this type on the summer high-Arctic aerosol using improved techniques and studying a remarkably large number of individual particles. I strongly recommend it for publication, but contribute a few comments that might be useful.

1. Sizing of the aerosol from electron microscope imagery is notoriously difficult and in this case has resulted in acceptable agreement with TDMPS size distributions. Subsequent investigators will want to try to compare their work with that under review. It should therefore be made clear what assumptions were made in sizing particles such as A in figure 4 where no shadow is visible, or B in that figure which shows a thin shadow on the lower edge suggesting that it is a flat crystal. Similarly, in figure 7, were the components of the chain aggregates assumed to be spherical, their volumes summed and the diameter of a sphere with the equivalent diameter calculated? If so, attention should be called to the paper by Rogak et al. (Aerosol Sci. Tech 18, 25-47, 1993) which showed that a mobility analyser bases diameter of such particles on the projected area rather than on the volume. This will affect the comparison of mobility and EM size distributions.
We are aware of this problem and therefore used the projected area of a particle to determine the particle equivalent diameter $D_{pa}$ (the diameter of a circle that comprises the same diameter as the projected particle), see chapter 2.2.4.

2. P.10 line 24. I can't understand why C and O were not detected on blank films of polyvinyl formal. In the supplemental data the carbon signature is strong, so the detector was sufficiently sensitive.
Thank you for pointing out this error. We changed the text to:
P10 lines 23-25. "The EDX spectra of blank grids showed only signals from Pt, the supporting copper TEM grid and carbon and oxygen signals from the Formvar substrate film."

3. Droplet haloes: I don't believe the splash hypothesis is appropriate for low velocity electrostatic collection. In fact I think it is also doubtful for particles of the size of those in figure 10 collected by high velocity impaction. Stratospheric aerosol sampling by an Ames Research Center group (Farlow and Ferry) 40 years ago found that the sulfuric acid particles did not develop haloes if all contact with water vapour was avoided before examination. (Possibly in JGR 82, 4921-4929, 1977 but I don't have the article) It was later confirmed by laboratory experiments (Bigg, Tellus, 38B, 62-66, 1986). A possible explanation is that acid vapour extends outwards from the captured particle as a monolayer (or multiple layers) on the surface. On exposure to water vapour the molecules take up water and coalesce to form tiny droplets.

We agree, halo formation has to be discussed as the result of several factors. We therefore extended the discussion of halo formation as follows:
P16, lines 5-15: "Several authors (Farlow et al., 1977; Bigg, 1986; Bigg and Leck, 2001b) have found that the formation of droplet ring structures from sulphuric acid containing aerosol is a result of humidity, hydrophilicity of the collection surface and impact velocity effects. Bigg and Leck (2001b) observed that a solution of sulfuric acid wets out on a hydrophilic surface but retracts when humidity is reduced, leaving behind small droplets in a symmetrical ring. In our study the sampling procedure led to a drastical reduction in relative humidity, from around 100% at ambient conditions to 20% within the sampling manifold (see chapter 2.1.1) and the aerosol was impacted onto a surface with hydrophilic properties (TEM grid). We will thus assume that the HP originally existed as one particle in the atmosphere that splashed out into the droplet ring structure upon impaction onto the substrate. "

4. P.15, line 23: "the biopolymer networks of marine gels are water solvable". Solvable means that an answer is available for a problem. If you meant "soluble", how could they exist as entities in the ocean?

Thank you for pointing that out. We now use "solvated" instead, referring to the IUPAC definition for solvation: "Any stabilizing interaction of a solute (or solute moiety) and the solvent or a similar interaction of solvent with groups of an insoluble material (i.e. the ionic groups of an ion-exchange resin)…"

We changed the text to:
P15, line 17-31: "In seawater the observed size range of gel particles ranges from sovated nanogels (100-200 nm; Bigg et al., 2004) that can further anneal into microgels (> 1000 nm) by interpenetration and entanglement of neighboring nanogels or hydrophobic interaction."

The manuscript is well-written, the diagrams informative and the references very comprehensive. There are some instances where the spelling or wording differs slightly from conventional English usage and some of these are listed below together with suggested alternatives.

Thank you for thoroughly reading our manuscript. We corrected spelling and wording as suggested below.

p.5, line 21: Longyearbyen, Svalbard

p.5, line 25 and beyond: Since you are reporting completed work it would be more conventional to use the past tense rather than the future tense. E.g., change "will use" to "used".

p.6, line 2: according to morphological. . .

p.8, line 19: In order to compare (to) the number. . .

p.9, line 21: Although wolfram is more logical in view of its symbol, tungsten is the common English usage.

p.10, line 19: were not reliably detected

p.16 line 10: unstable

p.16 line 17: morphology to the

p.20 line 13: is capable of adding

p.20 line 30: In the hope of enhancing

References, p.23, line 5: Ayers, G.P.

p.23 line 8: pouchetii

p.25 line 4: Cambridge

p.30 line 21- 24: Remove hyphens in Ramaswamy, Isaksen, climate and Intergovern-
mental

p.31, line 6: atmosphere

Figure 10 caption, line 6: degenerated

---

## Author Comment (AC4) · 11 Apr 2016

[Figure]

Figure S1: Examples of EDX spectra of aerosol particle investigated in this study. (A) single particle containing neither $Na^+/K^+$ nor $Ca^{2+}/Mg^{2+}$. (B) *"aggregate"* particle containing $Na^+/K^+$. (C) *"aggregate with film"* particle containing $Na^+$ and minor contents of $Ca^{2+}/Mg^{2+}$. (D) *"mucus-like"* particle containing $Ca^{2+}/Mg^{2+}$ and minor contents of $Na^+$. (E) halo particle containing $Na^+/K^+$ and S probably in the satellite particles. (F) blank spectrum taken from a Pt-shadowed TEM grid.

---

## Author Response (AR2)

Dear Prof. Carslaw,

On behalf of all authors I would like to thank you again for taking the time to edit our manuscript.

As requested a native English speaker looked over the manuscript. The following major changes have been made (marked in red in the manuscript below):

The sentence starting on p.6, line 15 has been rephrased.
The sentence starting on p.12, line 24 has been rephrased.
The sentence starting on p.13, line 13 has been rephrased.
The sentences starting on p.17, line 30 to p.18, line 2 have been rephrased.

The figure captures have been revised.

The paragraph starting on p.16, line 17 has been rewritten in order to clarify that the fragmentation mechanisms in the atmosphere are not well established. Part of the final sentence of this paragraph has been moved to the footnote on p.5.

We hope that the quality of writing in the manuscript has improved and is sufficient for publication in ACP now.

Yours sincerely,
Evelyne Hamacher-Barth

[revised manuscript text omitted]